# OpenReview forum: "Benchmarking Mitigations For Covert Misuse"
_ICLR.cc/2026/Conference — Submitted to ICLR 2026_

### Official Review · Reviewer_fhmj · 2025-10-18

**Soundness:** 3
**Presentation:** 3
**Contribution:** 3
**Rating:** 6
**Confidence:** 3

**Summary:**

This paper addresses the limitation of existing LLM safety evaluations that focus only on overt misuse, by systematically studying covert decomposition attacks—where attackers split a harmful goal into multiple benign-looking sub-queries to evade refusal and still achieve malicious outcomes. The authors introduce BSD (Benchmarks for Stateful Defenses), a new evaluation framework that automatically generates biosecurity and cybersecurity questions which are difficult, consistently refused by safety-trained models, yet solvable by unaligned models. BSD enables measurement of both misuse uplift (how much strong models inadvertently help attackers) and detectability.

**Strengths:**

The paper tackles an underexplored yet realistic threat, covert multi-query misuse, shifting the focus from single-prompt safety to stateful, multi-turn risk assessment.

The proposed BSD pipeline systematically generates difficult, refused, and answerable misuse questions, filling a major gap in current LLM safety evaluation.

**Weaknesses:**

The BSD dataset is not publicly released, which significantly hinders reproducibility and independent validation of the results.

The evaluation focuses mainly on biosecurity and cybersecurity; other plausible misuse domains such as misinformation or social manipulation remain untested, limiting generalizability.

The study does not deeply examine adaptive attacker strategies (e.g., cross-account evasion, prompt obfuscation).

**Questions:**

How do you ensure that the BSD-generated tasks remain representative of real-world covert misuse scenarios rather than artificial examples tailored to model refusal behavior?

Have you tested whether decomposition attacks still succeed when the “strong” and “weak” models are replaced with different architectures or model families (e.g., Claude vs Gemini)?

Could adaptive attackers deliberately design sub-queries to mimic benign user behavior or cross multiple accounts to evade the buffer defense, and how might you detect this?

---

> ### Author Response · Authors · 2025-11-24
> **Rebuttal part 1**
>
> We thank the reviewer for their thoughtful questions. We consider their points below.
>
> ## Weaknesses
>
> >The BSD dataset is not publicly released.
>
>
> Due to concerns around misuse, we unfortunately cannot release our full dataset. We nonetheless have a short open access form where we have already shared the dataset with a number of academics, private frontier lab employees, and safety institutes. Similarly, we have been very responsive with the groups who also have reproduced our results (in unpublished work). Finally, to make the release policy more transparent, we attach our questionnaire we send with the form in a new appendix item (Appendix I).
>
>
> > [...] other plausible misuse domains such as misinformation or social manipulation remain untested, limiting generalizability.
>
> We agree that our datasets do not cover all the possible misuse directions, and we hope to generalize this in future work. In the current work, we choose to focus on domains where models may soon be able to help actors commit significant harm, and therefore deserve focus. Our current work is effective at measuring the risks of decomposition attacks on models with frontier safety training such as GPT-5, as our updated table shows. It also provides sufficient samples for us to be the first work to evaluate stateful defenses in Section 5. We view our work as a significant step towards more realistic attack and defense evaluations, and a necessary direction for the future of AI safety research.
>
> >The study does not deeply examine adaptive attacker strategies.
>
> We want to highlight that (1) we consider adaptive attack strategies to overflow the stateful defense buffer in Section 5 of our paper and (2) look at prompt obfuscation/jailbreaking in Appendix G, where we compare the effectiveness of decomposition attacks, GCG, and their combination on Llama-3.1 to show that our method is even more effective than the scenario where the guardrails of the model are erased. We have added a paragraph to explain this in Section 6. As for black-box models, where GCG is not directly applicable, we combined decomposition attacks with PAIR against Claude-3.5 and GPT-5. However, this did not improve the results since the sub-tasks were either still refused, or in case that they were answered, they did not lead to a correct answer by the composer. We have detailed this in the response to reviewer jAQG, and can add a more comprehensive discussion in appendix G.
>
> ## Questions
>
> > How do you ensure that the BSD-generated tasks remain representative of real-world covert misuse scenarios [...]?
>
> This is a great question. We point to two lines of evidence. First, we note that our pipeline generates questions that are consistently agreed upon by frontier models (a strong proxy for correctness [1]). These questions are also consistently refused: that is, the questions are harmful “enough” in that they are robustly refused by different policies across models; this suggests that these tasks are not just finding edge cases in a single model’s refusal policies.
>
> The second line of evidence we use to check for the representativeness of the BSD tasks is qualitative examination. While we do not conduct a full human-study due to misuse concerns, we did allow cyber and bio experts to examine our tasks to confirm correctness. No problems were found. The authors of the paper also manually filtered the question sets via (1) extensive literature review and google searching to check for factual correctness and (2) filtering the tasks for diversity, to enforce that the final set of tasks covered a wide range of topics.
>
> [1] Vendrow, Joshua, et al. "Do large language model benchmarks test reliability?." arXiv preprint arXiv:2502.03461 (2025).
>
> > Have you tested whether decomposition attacks still succeed when the “strong” and “weak” models are replaced with different architectures.
>
> We have included two new models (Gemini 2.5 Pro and GPT-5) in Table 1 per the request of the reviewer, where the decomposition attack performs notably well against GPT-5. We hope that the results are convincing around the effectiveness of our pipeline and the attack. GPT-5 nearly refuses all of the questions yet answers to 50% of sub-tasks.

---

> > ### Author Response · Authors · 2025-11-24
> > **Rebuttal part 2**
> >
> > > Could adaptive attackers deliberately design sub-queries to mimic benign user behavior or cross multiple accounts to evade the buffer defense, and how might you detect this?
> >
> > The reviewer is correct, and at the moment a very determined attacker could likely distribute queries across many accounts (e.g. potentially created with different IP addresses so that they are hard to trace). However, this is practically difficult, as creating new accounts adds significant overhead: it requires creating new email addresses and new banking/credit card numbers (the latter of which is particularly difficult due to “know-your-customer” requirements in banking). Indeed, in the recent well-publicized decomposition attack from (purportedly) a sophisticated nation state threat actor mentioned in our response above, each decomposition campaign seemed to come from a single account [1]. Nonetheless, we think that our results, and attacks like the ones mentioned in our paper and [1], might suggest model providers should implement stronger measures on knowing their customer in order to prevent such misuse.
> >
> >
> > [1] https://www.anthropic.com/news/disrupting-AI-espionage

---

### Official Review · Reviewer_EwCY · 2025-11-01

**Soundness:** 2
**Presentation:** 2
**Contribution:** 2
**Rating:** 2
**Confidence:** 4

**Summary:**

1. The paper introduces Benchmarks for Stateful Defenses (BSD), a pipeline to evaluate model safety against decomposition attacks, where an attacker breaks a harmful task into multiple benign queries.

2. The paper argues that existing single-prompt defenses are insufficient and highlight stateful defenses, which keep track of a user's query history, as a necessary and promising countermeasure.

**Strengths:**

1. The paper clearly explains the concept of misuse uplift, which is the incremental advantage a strong, guarded model provides over a weak, unsafe model when solving a harmful task.

2. The authors present an automated data generation pipeline (BSD) that successfully curates datasets in biosecurity and cybersecurity, filtered to be difficult for weak models and answerable but consistently refused by strong, safe models.

3. The paper evaluates defenses, showing that adversarially training a prompt-level detector offers some benefit, and proposes a novel stateful buffer defense that tracks suspicious queries over time.

**Weaknesses:**

1. The final curated benchmark only consists of 50 biology questions and 15 cybersecurity questions. This limited size may not be comprehensive enough to draw broad conclusions about model safety.

2. ⁠The entire decomposition attack chain relies on a critical assumption: that a "weak, unsafe model W" (e.g., Qwen2.5-7B) is capable of intelligently decomposing a difficult, harmful task (X) that it cannot solve itself. This decomposition is itself a highly advanced planning and reasoning task. The paper does not sufficiently justify why a model that is "weak" at solving a problem would simultaneously be "strong" at planning its decomposition. If the weak model's decomposition quality is poor, the entire attack chain would fail.

3. I think task decomposition is nothing new in LLM literature. This makes the paper's novelty very limited.

**Questions:**

See weaknesses and here are some additional questions:

1. Figure 3 indicates that 2815 generated questions were filtered out because they were not refused by strong, safe models. Does this high number of answered harmful questions itself represent a significant finding?

2. While state-based defences can protect against a user who decomposes a task into many individual queries, could an attacker simply bypass this defense by distributing their decomposed queries across multiple, hard-to-link accounts. Does this limit the defense's real-world viability?

---

> ### Author Response · Authors · 2025-11-24
> **Rebuttal part 1**
>
> We thank the reviewer for their useful and relevant comments. We address them below.
>
> ### Weaknesses
> >  The final curated benchmark only consists of 50 biology questions and 15 cybersecurity questions. This limited size may not be comprehensive enough to draw broad conclusions about model safety.
>
> We want to highlight that our generation pipeline can generate many more high-quality samples as our analysis shows, and are limited by budget constraints. We thereby view our pipeline and C1-3 in Section 3 as the main contribution. Nevertheless, the current BSD datasets with 50/15 examples yields hundreds of decompositions, due to the fact that these are difficult tasks with many sub-parts, with which we are the first to study stateful defenses and misuse uplift. With our dataset of difficult and harmful misuse questions and decompositions, we argue that we in fact can make strong conclusions around the need for stateful defenses for LLM safety due to the relative inadequacy of pointwise defenses to defend against decomposition attacks in Section 5. To our knowledge, we are the first to analyze these phenomena at this granularity, and our curated questions and decompositions are sufficient to draw strong conclusions—our significant results on Claude-3.5 and GPT-5—about the need for stateful defenses for LLM safety.
>
> We would also like to note that 50 example questions are on the same order of comparable safety evaluations from previous work [1, 2]. Furthermore, some of our most significant results on Claude and GPT-5 critically rely on conditions C1-3. These conditions impose tight constraints on the efficiency that makes data collection more costly, yet necessary for a realistic safety evaluation.
>
> [1] Mazeika, Mantas, et al. "Harmbench: A standardized evaluation framework for automated red teaming and robust refusal." arXiv preprint arXiv:2402.04249 (2024).
>
> [2] Zhang, Andy K., et al. "BountyBench: Dollar Impact of AI Agent Attackers and Defenders on Real-World Cybersecurity Systems." arXiv preprint arXiv:2505.15216 (2025).
>
> > ⁠The entire decomposition attack chain relies on a critical assumption […]
>
> We agree that this is an important detail, and if an attacker model is too weak, then it lacks the planning ability to implement a useful decomposition. Indeed, one of the central empirical claims in our paper backs this exact assumption: we do in-fact show that weak models can simultaneously struggle with a difficult misuse dataset (our BSD dataset) while still implementing successful decompositions. This is what our “misuse uplift” metric captures: in words, it tells us how useful the decomposition attack plan is beyond simply solving the problem with the weak model. One intuition that might be useful (and which we can put in the paper, if the reviewer agrees with its usefulness) is that certain classes of misuse questions are much easier to decompose than to answer directly. For example, a cybersecurity exploit task nearly always has natural subquestions (e.g., network probing, fingerprinting the exposed services, enumerating vulnerabilities, and proposing attack paths) that are easy to state but difficult to implement in scripting and code. We will discuss some realistic examples of such attacks in the following paragraph.
>
> Finally, we note that our threat model is meant as a proxy for bad actors who are willing to misuse available models, as motivated by the Las Vegas terror example in the introduction. Throughout the paper, we aim to motivate that a realistic attacker will not necessarily use the capabilities of frontier models for the entire planning process, but to get the answers to more granular and benign-looking questions. For example, this perspective is well-aligned with Anthropic’s recent report on the “very first orchestrated cyber espionage” case, where the attackers “broke down their attacks into small, seemingly innocent tasks that Claude would execute without being provided the full context of their malicious purpose” (link below). Our work is, to our knowledge, the first attempt to systematically benchmark such decomposition-based attacks and also the first paper to benchmark stateful defenses to such attacks. We hope the reviewer sees the value of this threat model for the future of AI safety, where defenses against decomposition attacks will be of paramount importance.
>
> Link to the report mentioned above: https://www.anthropic.com/news/disrupting-AI-espionage

---

> ### Author Response · Authors · 2025-11-24
> **Rebuttal part 2**
>
> >I think task decomposition is nothing new in LLM literature.
>
> We agree with the reviewer in that decomposition attacks were introduced in previous work. However, as described in the introduction, our contribution is to both rigorously benchmark, and introduce new defenses against, decomposition-based misuse in a realistic safety setting. We construct the first evaluation for defenses against decomposition attacks. To do this, we describe a new threat model around misuse uplift and detectability that motivates decomposition attacks. To test this threat model, we introduce a data pipeline that creates hard and refused questions. We show that decomposition attacks achieve higher uplift– and are harder to detect– than the best jailbreaks on the dataset created from our pipeline. With our evaluation, we propose and benchmark the first online stateful defenses for attacks on language models.
>
> We believe that while previous works motivated decomposition attacks, they were not properly benchmarked: [1] focuses on some synthetic coding tasks—which are not released—and, as shown in their Table 1, these are relatively feasible for the weak model itself, making them less representative of the “weak attacker / strong model” regime we focus on. [2] studies the decomposition attack through the lenses of information leakage on tasks that are answered (i.e. are not refused) by current frontier models, and does not report end-to-end success rates on actual difficult/harmful misuse tasks.
>
> Moreover, we deliberately did not list “a new decomposition attack” among our main contributions (even though we consider it a minor contribution), since we wanted to emphasize that we focus on evaluating them more realistically through our novel pipeline. Then, we proposed the first effective stateful defense specifically targeted at such attacks, which we show can be further improved by future work. Lastly, we want to mention that we had to implement the decomposition attacks from scratch and only from the high-level description in their paper, as there is no released code or dataset. This underscores the current lack of benchmarks in this space.
>
> [1] Jones et al., “Adversaries can misuse combinations of safe models.”
>
> [2] Glukhov et al., “Breach by a thousand leaks: Unsafe information leakage in ‘safe’ ai responses.”
>
> ## Questions:
>
> > Does this high number of answered harmful questions itself represent a significant finding?
>
> We agree with the reviewer, and think there are two relevant findings from our pipeline experiments worth noting on: (1) that GPT-4.1 is a relatively unsafe model, in that it is willing to generate many thousands of potential cyber and bio misuse tasks, and (2) as the reviewer noted, that models are generally willing answer such questions, and only do not answer the most extremely harmful misuse questions. We note that (2) holds for questions generated by models other than GPT-4.1 (namely, Qwen and DeepSeek models that are also largely unsafe), so we do not think this observation is model-dependent. Instead, we think that this speaks to the general policies that model providers take towards dual use information. We add discussion of this finding to the bottom of figure 4.
>
> >[...] could an attacker simply bypass this defense by distributing their decomposed queries across multiple, hard-to-link accounts?
>
>
> The reviewer is correct, and at the moment a very determined attacker could likely distribute queries across many accounts (e.g. potentially created with different IP addresses so that they are hard to trace). However, this is practically difficult, as creating new accounts adds significant overhead: it requires creating new email addresses and new banking/credit card numbers (the latter of which is particularly difficult due to “know-your-customer” requirements in banking). Indeed, in the recent well-publicized decomposition attack from (purportedly) a sophisticated nation state threat actor mentioned in our response above, each decomposition campaign seemed to come from a single account [3].
>
> [3] https://www.anthropic.com/news/disrupting-AI-espionage

---

### Official Review · Reviewer_jAQG · 2025-11-06

**Soundness:** 2
**Presentation:** 3
**Contribution:** 3
**Rating:** 4
**Confidence:** 5

**Summary:**

The paper presents a benchmark to address a critical gap in AI safety evaluations: existing benchmarks fail to capture realistic misuse patterns where attackers can decompose harmful tasks into benign-seeming queries (decomposition attacks). The authors introduce BSD (Benchmarks for Stateful Defenses), a dataset generation pipeline that produces 50 biology and 15 cybersecurity questions that are (1) difficult for weak models, (2) refused by safety-trained models, and (3) answerable by helpful-only strong models. They evaluate decomposition attacks and find that such achieve significant misuse uplift while evading prompt-level detection. The paper proposes stateful defenses that maintain buffers of suspicious queries across user sessions, showing promise but remaining imperfect against adaptive attacks.

**Strengths:**

- The threat model is well-motivated by actual incidents. The observation that attackers use benign-looking queries in practice, not direct harmful requests, is crucial and underexplored.
- Comparing decomposition with 5 other attack types (Adaptive, PAIR, Adversarial Reasoning, Crescendo, improved decomposition) under equal query budgets (12 calls) is methodologically sound and provides useful baselines.
- The finding that decomposition attacks are much harder to detect (AUC 0.60) than jailbreaks (AUC 0.71) even with adversarial training is significant and actionable for defenders.

**Weaknesses:**

- Since the uplift is defined as difference in accuracy between a weak and a strong model, and Qwen2.5-7B is used as the weak models in most experiments in the paper, the reported "uplift" numbers are inflated. For clarity in the paper, the metric should be called "relative uplift" so that it is clear to a reader who is skimming that the uplift numbers are relative to Qwen2.5-7B. Reporting the relative uplift numbers against multiple open-weights models would also be insightful.
-  The paper states in Section 4.1 that "each attack is run for five epochs" but still never reports variance. Table 1 doesn't have any confidence intervals. Figure 7 (left) also misses error bars which makes it difficult to refute that the scaling trend is within noise (no significance tests). This is particularly problematic since the benchmark has only 50 questions which could result in significant variance.
- Some of the defender assumptions in the threat model are not validated
    - Assumes defender can reliably track users across sessions (trivial to circumvent with multiple accounts/IPs)
    - No analysis of false positive rates on benign users doing legitimate multi-step research
- Figure 3 flow diagram is confusing (numbers don't clearly show what each filter removes)
- Heavy appendix reliance (key results like Appendix G relegated to supplementary)
- "Restricted release" of dataset is understandable but limits reproducibility.

Weak baseline and missing error bars and significance tests leads me to weak reject the paper in the current form. Addressing these should make the paper stronger.

**Questions:**

- What do intermediate helpful-only models score on BSD? Please add a table showing direct query accuracy for:
  - Qwen2.5-7B (current baseline)
  - Qwen2.5-72B (no safety training)
  - Llama-3.1-405B (no safety training)
  - Any 400B+ uncensored model you can access

  This is essential to validate that uplift is real and not inflated by weak baseline choice.
- Why not test decomposition + jailbreak hybrid in main paper? Appendix G shows this is effective (87% vs 84%) - this seems like the realistic threat model.
- What fraction of legitimate users doing multi-step research get flagged by your stateful defense at various thresholds?
- Have biosecurity or cybersecurity experts validated that your questions are genuinely harmful and your generated answers would facilitate misuse? A dataset that claims to measure uplift should consult with experts on this.

---

> ### Author Response · Authors · 2025-11-24
> **Rebuttal part 1**
>
> We thank the reviewer for their in-depth review and insightful comments, which resulted in uplift experiments on near-frontier open-weight models (Table 3 in App. F.1), an updated Table 1 with strong decomposition attack results, and more details around the stateful defense. We address the reviewers points below.
>
> ## Weaknesses:
> >  [...] metric should be called "relative uplift" so that it is clear to a reader who is skimming that the uplift numbers are relative to Qwen2.5-7B. Reporting the relative uplift numbers against multiple open-weights models would also be insightful.
>
> We agree that reporting uplift numbers for multiple models will be useful, and so include additional runs in appendix F.1 (Table 3) for Kimi K2, Qwen2.5-72B, Qwen3-235B, and Llama-3.1-405B. To add more to the clarity of our results, we have also included both the Qwen2.5 and direct query baselines to Table 1 in our revised version. However, we disagree with the characterizations that 1) the metric should be called “relative uplift” and 2) using Qwen2.5-7B “artificially” inflates uplift. Regarding the first point, “uplift” is already a unitless quantity; in standard usage, for example, “human uplift” describes the delta between a human with a language model versus a human without a language model. Of course, this uplift may be different for an expert human and an amateur human, nonetheless we think these are both quantities of interest. Similarly, while we think reporting uplift numbers is informative, we also think the current setting we consider holds up well: we are able to measure the effectiveness of decomposition attacks relative to jailbreaks for multiple different relative strengths of models, and also are able for the first time to compare the effectiveness of stateful vs prompt-wise defenses.
>
> >  Table 1 doesn't have any confidence intervals.
>
> We agree with this point, and have added the standard deviations to all of the experiments in Table 1. The updated table shows that, even in a limited setting with 50 samples, we get significant results especially when it comes to safer models such as GPT-5.
>
> > Some of the defender assumptions in the threat model are not validated [...] Assumes defender can reliably track users [...] No analysis of false positive rates.
>
> Regarding the first point, we agree that an attacker can in theory distribute queries across many accounts (e.g. potentially created with different IP addresses so that they are hard to trace). However, this is practically difficult, as creating new accounts adds significant overhead: it requires creating new email addresses and new banking/credit card numbers (the latter of which is particularly difficult due to “know-your-customer” requirements in banking). Indeed, in a recent well-publicized decomposition attack from (purportedly) a sophisticated nation state threat actor mentioned in our response to reviewer wohN, each decomposition campaign seemed to come from a single account (although there were multiple account bans, due to there being different campaigns) [1]. Nonetheless, we think that our results, and attacks like the ones mentioned in our paper and [1], might suggest model providers should implement stronger measures on knowing their customer in order to prevent such misuse.
>
> Regarding the second point on analysis of false positive rates, while Figure 6 in-principle provides this information, we agree that specific information is hard to pull out. In Table 5 (Appendix F.4), we now provide the precision and false positive rate for the pointwise defense and our buffer defense for recalls of 90% and 99% respectively. This is the setting where a “benign” user is asking for WMDP tasks and decompositions, so the false positive rates here reflect “benign users doing legitimate multi-step research.” The buffer defense dramatically outperforms the pointwise defense in realistic attack settings, and maintains similar precision values even when the signal is very noisy, and the harmful prompts are rare with respect to the benign prompts.
>
> [1] https://www.anthropic.com/news/disrupting-AI-espionage
>
> >Figure 3 flow diagram is confusing (numbers don't clearly show what each filter removes)
>
> We agree with the reviewer that Figure 3 needed clarification. In this regard, we have added more explanation to the caption in the revised version.

---

> ### Author Response · Authors · 2025-11-24
> **Rebuttal part 2**
>
> >Heavy appendix reliance (key results like Appendix G relegated to supplementary)
>
> We agree that applying jailbreaking methods on top of the sub-queries is an important/novel technical direction and merits discussion in the main body. We have added a new paragraph in Section 6 that summarizes Appendix G.
> For the remaining appendices, we want to note that our reliance on the appendices is largely driven by space limitations and the breadth of the contributions. We understand that this can be frustrating for the reviewers, but we nonetheless try to defend all of our core claims with results located in the main body. Again, we thank the reviewer for this point, especially regarding Appendix G.
>
> >"Restricted release" of dataset is understandable but limits reproducibility.
>
> We are glad the reviewer appreciates the safety concerns with our dataset release– we would like to note that, by construction, our dataset questions are significantly more harmful and prone to misuse than existing published datasets. Due to this, along with the bio and cyber experts we consulted, we decided that our release policy was necessary. There is a dataset access form linked in our non-anonymized paper, and we have already shared the dataset with a number of academics, private frontier lab employees, and safety institutes. Similarly, we have been very responsive with the groups who also have reproduced our results (in unpublished work). To make the release policy more transparent, we attach the questionnaire we send with the form in a new appendix item (Appendix I).
>
>
> ### Questions:
>
> > What do intermediate helpful-only models score on BSD?
>
> Great question. We have added this information for Kimi K2, Qwen2.5-72B, Qwen3-235B, and Llama-3.1-405B in what is now Table 3, in section F.1. We also include results from implementing decomposition attacks with these models against GPT-5 and Gemini 2.5 Pro. As this section now discusses, the misuse uplift for these models is (unsurprisingly) more modest compared to the weaker Qwen 2.5-7B, but still significant. We thank the reviewer for suggesting this experiment.
>
> >Why not test decomposition + jailbreak hybrid in the main paper? Appendix G shows this is effective (87% vs 84%) - this seems like the realistic threat model.
>
> We think more experiments are needed before we can present a complete message on the decomposition + jailbreak hybrid setting in the main body. As noted in lines 300–302 of the revised version, GPT-5 already answers 48.3% of the sub-queries; although this rate can potentially be improved with jailbreaks such as GCG, we are doubtful about the full efficacy of this idea. For less safe open-source models such as Llama-3.1, our observation is that jailbreaking helps bypass some refused sub-queries. However, for GPT-5 and Claude-3.5, which are black-box models (so GCG is not directly applicable), we experimented with PAIR on top of the refused sub-queries and did not observe any significant improvement. There were two main failure modes: (1) a portion of the sub-queries were still refused, and (2) when they were answered, the additional responses did not  help with finding the correct answer for the composer. This is consistent with recent work [1] that shows that jailbreaking impacts the quality of the answer. With careful tuning, we suspect that decompositions with jailbreaking will outperform decompositions, however we think the scale of this improvement will be, for example, significantly less than that of our improved decomposition attack. We will add this clarification to Appendix G.
>
> [1] Nikolic et al., “The Jailbreak Tax: How Useful are Your Jailbreak Outputs?”
>
> >What fraction of legitimate users doing multi-step research get flagged by your stateful defense at various thresholds?
>
> Thank you for this question– while Figure 6 in-principle provides this information, we agree that specific information is hard to pull out. In Table 5 (Appendix F.4), we now provide the precision and false positive rate for the pointwise defense and our buffer defense for recalls of 90% and 99% respectively. As discussed, the buffer defense dramatically outperforms the pointwise defense, maintaining similar precision values even when the signal is very noisy, and the harmful prompts are rare with respect to the benign prompts.
>
>
> > Have biosecurity or cybersecurity experts validated that your questions are genuinely harmful and your generated answers would facilitate misuse?
>
>
> While we did not conduct a full human-study due to misuse concerns, we did conduct spot checks where cyber and bio experts examined our tasks to confirm correctness. No problems were found. The authors of the paper also manually filtered the question sets via (1) extensive literature review and google searching to check for factual correctness and (2) filtering the tasks for diversity, to enforce that the final set of tasks covered a wide range of topics.

---

### Official Review · Reviewer_wohN · 2025-11-07

**Soundness:** 2
**Presentation:** 2
**Contribution:** 3
**Rating:** 6
**Confidence:** 3

**Summary:**

This paper tackles an important gap in AI safety evaluations by introducing a framework to measure "misuse uplift" through decomposition attacks. The authors argue that existing safety benchmarks are too easy and don't capture real-world threat scenarios where attackers break harmful tasks into benign-looking queries. They develop BSD (Benchmarks for Stateful Defenses), a data generation pipeline producing questions that are difficult for weak models yet consistently refused by frontier models. The evaluation reveals that decomposition attacks can significantly increase misuse rates while evading prompt-level defenses, and proposes stateful defenses that track user query histories as a countermeasure.
The core contribution is formalizing the misuse uplift concept (Δ = r_attack - r_weak) and creating evaluation infrastructure that properly measures it. The paper includes extensive experiments across multiple frontier models, various attack methods, and defense mechanisms.

**Strengths:**

1. The threat model is grounded in reality. The Las Vegas attack example and the employment fraud case study effectively motivate why we need to move beyond simple refusal-based evaluations. The distinction between "can this model do harm" versus "does this model provide incremental advantage over alternatives" is genuinely useful for thinking about deployment risks.
2. The filtering pipeline is well designed as it requires unanimous agreement from multiple strong models for ground truth, combined with consistent refusal from safety-trained models and difficulty for weak models, creates questions that meet clear criteria. The correlation analysis showing BSD performance tracks biological reasoning ability (ρ=0.94) while WMDP doesn't (ρ=0.11) is convincing evidence the questions measure genuine capability rather than memorized facts.
3. The stateful defense direction is novel for LLMs. While stateful defenses exist in computer vision, adapting them to track suspicious patterns across user sessions is an interesting contribution. The buffer-based approach that maintains the top-m most suspicious queries is practical and shows measurable improvements over naive rolling windows.

**Weaknesses:**

1. The decomposition attack improvements feel incremental. Section 6 presents fine-tuning the decomposer on 700 benign MMLU examples as a major contribution, but the gains are modest (Table 1). For instance, on Claude 3.5 the improvement is 41.6% to 46.0%, and the ablation in Table 3 shows minimal difference when no strong model is involved. This suggests the technique is more about optimizing prompt engineering than discovering a fundamental new attack vector. The claim that this represents "state-of-the-art" performance needs more justification given the limited scope.
2. Dataset release strategy undermines reproducibility. I understand the safety concerns, but committing to "restricted release under controlled access only" without specifying concrete access criteria makes it difficult to assess how useful BSD will be for the research community. The paper would benefit from either a clearer access policy or a compelling argument for why this approach is necessary when WMDP, which faces similar concerns, has been publicly released. The cybersecurity subset has only 15 questions, which seems insufficient for drawing strong conclusions about that domain.
3. Stateful defense evaluation lacks key details. Section 5.2 introduces the buffer defense but several important aspects remain unclear. How is the "harmfulness score" for individual prompts computed? The paper mentions using "adversarially trained Llama-Guard" but doesn't specify whether this uses the same 20% train split or different data. The choice of m=30 for buffer size appears arbitrary. Most critically, the false positive rate discussion is limited. Figure 6 shows precision dropping substantially as the fraction of harmful queries decreases, but there's no analysis of how this would affect legitimate power users who make many technical queries.
4. The weak/strong model distinction relies on fragile assumptions. Appendix B argues extensively that attackers will have access to both weak helpful-only models and strong safety-trained models. While currently plausible, this assumption could break down quickly. The paper acknowledges open-weight models are catching up (they cite DeepSeek and note the 6-month gap is closing) but doesn't adequately address how this affects the threat model's validity. If Qwen3-235B or similar models reach GPT-4 level performance, the entire misuse uplift framework becomes less meaningful because the attacker wouldn't need the strong model at all.
5. Statistical rigor is limited. The paper reports single accuracy numbers without confidence intervals or significance tests. Given that BSD has only 50 biology questions, and some models answer stochastically, uncertainty quantification would strengthen the claims. The paper mentions running decomposition "for five epochs" and "at least 4 out of 5 runs" but doesn't consistently report variance across runs. This makes it hard to assess whether observed differences are meaningful or within noise.
6. The relationship between detectability and utility is underexplored. The paper demonstrates that decomposition attacks are harder to detect than jailbreaks (Figure 5, AUC 0.60 vs 0.71) and that adversarial training helps (AUC increases to 0.76). However, there's no discussion of the tradeoff between detection accuracy and false positives in production settings. What happens to a legitimate bioinformatics researcher asking detailed questions about viral mechanisms? The precision-recall curves in Figure 6 show concerning drops at low base rates, but this isn't connected back to deployment recommendations.

**Questions:**

1. You filter down from 4800 candidates to 50 final questions (about 1% retention rate). This seems extremely aggressive. What are the most common failure modes? Do questions fail primarily at the "strong model agreement" stage, the "safety-trained model refusal" stage, or the "weak model difficulty" stage? Understanding the bottleneck would help assess whether the pipeline generalizes to other domains beyond biology and cyber.

2. The paper focuses on decomposition attacks because they're effective and hard to detect. But Section 4.1 shows that for some models (o3-mini, GPT-4o), traditional jailbreaks like adversarial reasoning achieve similar or better misuse rates (Table 1). Under what conditions should we expect decomposition to be the preferred attack strategy versus just investing more effort in jailbreaking? The paper would benefit from a decision tree or framework for when each attack class is most effective.

3. How does the buffer handle concept drift over time? If a user's interests legitimately shift toward more technical topics, could their older benign queries dilute the suspiciousness score? Conversely, what prevents an attacker from gaming the system by front-loading thousands of benign queries before attempting misuse? The adaptive attack in Section 5.2 touches on this but assumes random interleaving. A more strategic attacker might optimize the temporal pattern.

4. You compare against several jailbreak methods but not against other decomposition approaches. Glukhov et al. 2024 also proposes decomposition attacks. How does your improved decomposition method (fine-tuned decomposer + variable n) compare quantitatively to their approach on the same tasks? This would help establish whether the gains come from better evaluation infrastructure or genuinely better attacks.

5. The biology questions achieve strong results but cybersecurity shows more modest uplift (Figure 8). You attribute this to lower baseline refusal rates but don't deeply investigate. Are cybersecurity tasks fundamentally different in ways that make decomposition less effective? Or is this about BSD-cyber having easier questions? Testing the same pipeline on additional domains (chemistry, general engineering, persuasion) would clarify whether this is a principled evaluation framework or one tailored to specific biology hazards.

6. You fine-tune on MMLU-auxiliary decompositions from o3-mini with n varying from 3 to 12. Why this particular distribution? Did you experiment with other training data compositions (pure n=12, or weighted toward higher n)? Also, the ablation shows minimal improvement without a strong model, but have you verified the fine-tuned decomposer doesn't leak information about the answer by generating leading sub-questions?

7. The stateful defense requires tracking user history across sessions and maintaining buffers. What's the computational overhead? How does this scale to millions of users? More importantly, how do you handle shared accounts or VPNs where multiple distinct users might appear as one "user" to the tracking system? These practical concerns seem important for real deployment but aren't addressed.

---

> ### Author Response · Authors · 2025-11-24
> **Rebuttal part 1**
>
> We thank the reviewer for their detailed and insightful comments. We appreciate their notes around the realism of our threat model, and the strength/novelty of our pipeline and stateful defense. Below, we address the reviewer’s questions and listed weaknesses.
>
>
> ### Weaknesses
> > The decomposition attack improvements feel incremental.
>
> We agree with the reviewer that the new attack makes modest gains over existing attacks. However, we would emphasize three points around our contribution and novelty. First, we emphasize that this threat vector, where an adversary finetunes a model on easily obtained harmless reasoning data with the goal of making a stronger attacker model is, to our knowledge, novel. In fact, there is a concurrent ICLR 2026 submission where their main contribution is this precise threat model [1]. Second, we would like to note that we had to implement the task decomposition baseline from the scratch and only from the high-level description in their paper, as there were no published available implementations/dataset. We spent a great deal of time optimizing this new implementation, and strongly suspect, although we cannot confirm, that it is stronger than previous work. Third, as mentioned in line 454 (464 in the revised version), the point of the ablation studies and Figure 7 is to show that the fine-tuned model without access to a strong model does not perform better than the base model, and the performance gains are indeed because of the better decomposition quality.
>
> > Dataset release strategy undermines reproducibility. [...] The paper would benefit from either a clearer access policy [...] when WMDP, which faces similar concerns, has been publicly released.
>
> We are glad the reviewer appreciates the safety concerns with our dataset release– we would like to note that, by construction, our dataset questions are significantly more harmful and prone to misuse than WMDP questions (which are only adjacent to misuse). Due to this, along with the bio and cyber experts we consulted, we decided that our release policy was necessary. There is a dataset access form linked in our non-anonymized paper. To make the release policy more transparent, we attach the questionnaire we send with the form in a new appendix item (Appendix I).
>
> > The cybersecurity subset has only 15 questions, which seems insufficient for drawing strong conclusions about that domain.
>
> We note that this still yields on the order of hundreds of decompositions for use in our stateful defense setting. We would also like to note that this number of tasks are on the same order of safety evaluations with comparable difficulty or harmfulness in previous work [2, 3]. Furthermore, we note that budget constraints limit the size of our dataset. As shown in Section 3.1, the pipeline is capable of generating high-quality and diverse examples.
>
> > Stateful defense evaluation lacks key details.
>
> We now include more information around the defense performance in Appendix F.4; Table 5  shows the precision and false positive rate for the pointwise defense and our buffer defense for recalls of 90% and 99% respectively. As discussed, the buffer defense dramatically outperforms the pointwise defense, maintaining similar precision values even when the signal is very noisy, and the harmful prompts are rare with respect to the benign prompts.
>
> We discuss the choice of buffer size in Appendix F.2: we attempt to use as much effective context as possible, and because a realistic defender model may be unlikely to have a context length larger than 32K tokens, this accounts to only m = 30 queries, i.e. [\~30 prompts] x [\~1K tokens per prompt]. We will add this discussion to the main body of the paper if the reviewer thinks it is needed. Finally, as the reviewer noted, we use the adversarially trained Llama-Guard to compute the prompt-wise harmfulness scores; this model was trained with the 20% train split.

---

> > ### Author Response · Authors · 2025-11-24
> > **Rebuttal part 2**
> >
> > > The weak/strong model distinction relies on fragile assumptions.
> >
> > This threat model relies on the empirical observation that the gap between private and open-weights language models has continually persisted since 2022– in fact, this gap has likely re-widened in the period after the DeepSeek-R1 release [4]. In Appendix B.1 we nonetheless relax this assumption and argue that our misuse uplift threat model and stateful defenses will still be a useful setting for three reasons. We highlight them briefly here, and refer the reviewer to appendix B.1 and B.2 for further discussion:
> >
> >
> > (1) Even if open-weights models “catch-up,” private models might still have complementarities with open-weights models (because, for example, they have been trained on expensive biological laboratory experiments).
> >
> > (2) Open-weight LLM uplift is a proxy for human uplift, and we will likely want to know how bad actors may misuse a model even if open-weights models catch-up. In a similar vein, the defender’s private model may have better scaffolding (e.g. Claude Code vs open source alternatives) or better speed (companies can often secure access to faster state-of-the-art GPUs), and these can often aid in making misuse easier. In an example that motivates both of these points, we refer the reviewer to a very recent attack on Anthropic that uses the same threat:
> >
> > https://www.anthropic.com/news/disrupting-AI-espionage
> >
> >
> > Our work therefore motivates an automated benchmark for attacks and defenses that are both more realistic and effective than traditional jailbreak evaluations. Taken together, we are confident that the methods described in this paper will generalize to threat models with different assumptions around the weak/strong model distinction.
> >
> > > Statistical rigor is limited.
> >
> > In addition to the standard deviation already shown in Figure 1, we include the standard deviation over evaluation epochs for Table 1 in the updated version. We hope that this provides  more confidence in the reproducibility of our results.
> >
> > > The relationship between detectability and utility is underexplored. [...] The precision-recall curves in Figure 6 show concerning drops at low base rates, but this isn't connected back to deployment recommendations.
> >
> > We agree with the reviewer on the need to ground detectability in utility, and think our work, by by introducing the first actual evaluation for decomposition attacks and stateful defenses, is a relevant contribution towards in our work. First, as we note in our paper, the detection setting we consider is difficult by design: the defender is asked to differentiate between two distributions with very similar technical content (BSD vs. WMDP questions/decompositions), differing only in whether they are direct misuse questions. So, we think that our setting already captures the difficulty of the reviewer’s example: “What happens to a legitimate bioinformatics researcher asking detailed questions about viral mechanisms?”
> >
> > We agree that, due to the difficulty of differentiating misuse decompositions from benign questions that are technically related, there are no particularly desirable points on the operating curve that simultaneously achieve high recall on misuse and low false positives on legitimate users. This is why we are cautious in our message and instead position one of our main contributions as highlighting that current detectors struggle in this realistic regime.
> > Nonetheless, due to the fact that the buffer defense shows substantial improvements over prompt-wise defenses in Figure 6 (even during the adaptive attack!), we think our work strongly suggests to adapt such defense in practice. While the exact point the defender chooses on the utility-detectability operating curve will depend on the defender’s application and policy,  our work proposes two  concrete contributions: (i) benchmarking safeguards against decomposition attacks; and (ii) adopting stateful defenses. We would be happy to further explain the detection tradeoff and the implications in the revised paper.
> >
> >
> > ### Questions:
> > > You filter down from 4800 candidates to 50 final questions [...]. What are the most common failure modes?
> >
> > This is a great question! We aim to provide insight into this in the flow diagram in Figure 3: for bio, about 16% (758 filtered out of the initial 4800) are filtered due to correctness, about 70% of the remaining questions are filtered (2815 filtered) due not being sufficiently harmful, and then 95% of those remaining questions (1170 filtered) are filtered due to not being sufficiently difficult (in our figure, we also include the 7 questions we remove for being insufficiently diverse). So in general, the most likely reasons for a proposed question to be rejected is that it is either not sufficiently harmful or not sufficiently difficult, and a relatively smaller number get filtered due to not being correct.

---

> > > ### Author Response · Authors · 2025-11-24
> > > **Rebuttal part 3**
> > >
> > > > Under what conditions should we expect decomposition to be the preferred attack strategy [...]?
> > >
> > > We thank the reviewer for surfacing this. To better study this question, we have added an extra column in Table 1 that shows the direct query rates (i.e. simply prompting the model directly). We also conducted new experiments that show that the sub-queries (i.e., the subtasks from the attacker decomposition) have significantly higher compliance rates– are more likely to be answered– as they are designed to be benign-looking. In general, we think that decomposition attacks outperform others when there is a significant gap between the direct query and sub-query compliance rate.  Similarly, we think that attackers might prefer a decomposition attack over e.g. a jailbreak attack when they prioritize not being caught by the attacker. That is, jailbreaks often require many samples that get initially refused, whereas decomposition attack queries are “stealthier” in that individual queries are less commonly refused. We have updated the results paragraph in Section 4 to better explain this explanation and motivation, respectively. We will also add a new section in the appendix to show the compliance rate of decomposition queries across all models.
> > >
> > > >How does the buffer handle concept drift over time? [...] Conversely, what prevents an attacker from gaming the system by front-loading thousands of benign queries before attempting misuse?
> > >
> > > Regarding the first question: natural user concept drift is actually well-captured in our current evaluations, as we will explain. In our evaluations, we consider two distributions of questions / decompositions: BSD (which are refused misuse questions and are therefore labeled as “harmful”) and WMDP (which are in-policy/answered by frontier models, so we label them as “benign”). The WMDP questions and decompositions have the structure the reviewer asks about: they are both highly technical and cover diverse topics, and by construction are relatively similar to BSD queries. This is a difficult evaluation for the defense, and we find that the buffer defense works well in this setting.
> > >
> > > Similarly, regarding the second question (“what prevents an attacker from [...] front-loading thousands of benign queries?”), this is equivalent to the adaptive attack we consider against our buffer defense, for the following reason. The buffer is kept by ranking each query in a stream of queries by their isolated per-prompt harmfulness scores, and discarding those that do not meet some threshold (defined by some buffer size k). This combined operation of scoring, ranking, and then thresholding is invariant to the order of the queries– i.e., whether the queries are frontloaded or interleaved, so the front-loading adaptive attack is equivalent to our interspersing adaptive attack.
> > >
> > > > How does your improved decomposition method (fine-tuned decomposer + variable n) compare quantitatively to [Glukhov et al. 2024] on the same tasks?
> > >
> > > Unfortunately, [Glukhov et al. 2024] does not release their code, and as we noted, we had to recreate their decomposition attack (which is equivalent to the one in [Jones 2024]) ourselves from the high-level description in their paper. We spent a great deal of time optimizing this new implementation, and strongly suspect that it is stronger than previous work. Nevertheless, the reviewer can compare the quality of the examples provided in pages 21-23 with the examples given in previous work.
> > >
> > > > Are cybersecurity tasks fundamentally different in ways that make decomposition less effective? Or is this about BSD-cyber having easier questions?
> > >
> > > This is a great question– we are unsure, but suspect that the reviewer is correct, and BSD-Cyber has questions that are simultaneously easier and also less harmful (therefore more likely to be answered by frontier models). Both of these factors make misuse uplift smaller in this setting. However, our guess is that misuse uplift is actually still of great concern for cyber attacks, especially in agentic settings (whereas our setting mostly tests for reasoning and multi-step knowledge). For example, recently sophisticated attackers implemented cyber attacks with Claude by implementing what amounts to decomposition attacks: attackers “broke down their attacks into small, seemingly innocent tasks that Claude would execute without being provided the full context of their malicious purpose” [5].
> > >
> > >
> > > >Why this particular distribution for the number of decompositions?
> > >
> > > This choice was an attempt to avoid overfitting to any number of decompositions by the fine-tuned model; we initially tried fine-tuning with only n=3 that led the Qwen2.5-7B model to failure to generalize to other numbers. We did not try a different distribution over n, but we suspect that weighting towards larger n (or training only with n = 12) would only result in marginal improvements or degradations.

---

> ### Author Response · Authors · 2025-11-24
> **Rebuttal part 4**
>
> > have you verified the fine-tuned decomposer doesn't leak information.
>
> The most important factor for us to prevent any information leakage was fine-tuning the model only on MMLU-auxilary, which is orthogonal to our dataset. On page 23, we have provided examples of how the teacher model’s samples look on this dataset, which are irrelevant to our biology and biosecurity datasets. Furthermore, we think that any information leakage would impact the accuracy for the distilled model in Figure 7, and result in a meaningful jump in the accuracy, since it is both generating and answering the sub-queries in that experiment.
>
>
> >[...] What's the computational overhead? How does this scale to millions of users? More importantly, how do you handle shared accounts or VPNs where multiple distinct users might appear as one "user" to the tracking system?
>
> To first answer the final question around VPNs or shared accounts: we agree that determined attackers can attempt to conceal their account. However, this is practically difficult, as creating new accounts adds significant overhead: it requires creating new email addresses and new banking/credit card numbers (the latter of which is particularly difficult due to “know-your-customer” requirements in banking). We think our results around decomposition attacks, as well as recent real life attacks like those mentioned in our paper and [5], may suggest that providers implement measures to enforce even stronger links between users and accounts.
>
> To answer the initial questions: the buffer defense requires the classifier to incur a constant memory overhead for each user (e.g. a maximum of 30 prompts, or roughly [\~30 prompts] x [\~1K tokens per prompt] = 32K tokens. The cost of the forward pass is essentially the same as e.g. constitutional classifiers [6] (i.e., the current industry-standard point-wise defense). Fortunately, the classifier is relatively lightweight (e.g. an 8B model) and therefore the practical memory overhead is also modest.
>
>
>
> [1] https://openreview.net/forum?id=viBAbg9ihM
>
> [2] Mazeika, Mantas, et al. "Harmbench: A standardized evaluation framework for automated red teaming and robust refusal." arXiv preprint arXiv:2402.04249 (2024).
>
> [3] Zhang, Andy K., et al. "BountyBench: Dollar Impact of AI Agent Attackers and Defenders on Real-World Cybersecurity Systems." arXiv preprint arXiv:2505.15216 (2025).
>
> [4] https://evaluations.metr.org/deepseek-qwen-report/
>
> [5] https://www.anthropic.com/news/disrupting-AI-espionage
>
> [6] Sharma, Mrinank, et al. "Constitutional classifiers: Defending against universal jailbreaks across thousands of hours of red teaming." arXiv preprint arXiv:2501.18837 (2025).

---

### Author Response · Authors · 2025-12-03
**Summary of rebuttals**

We thank the reviewers and the AC for their engagement with our paper thus far, and are glad that all reviewers noted (1) the realism and usefulness of our new threat model (e.g., see the major attack discussed in a [recent threat report [1]](https://www.anthropic.com/news/disrupting-AI-espionage
)), (2) that our pipeline is able to show that decomposition attacks are concerningly effective misuse enablers,  and (3) the corresponding importance of our introduced stateful defense,

The reviewers who assigned lower scores had their concerns addressed. Reviewer EwCY (score of 2) based their critique on a misunderstanding of our threat model– they believed we had to assume weak models can decompose tasks they cannot solve, when in fact demonstrating this capability is precisely what our misuse uplift metric measures empirically. Reviewer jAQG (score of 4) requested several specific experiments, all of which we conducted: we added uplift measurements for four near-frontier open-weight models (Table 3) where decomposition still yields significant uplift, included standard deviations across Table 1, and provided detailed false positive analysis (Table 5).

**Concerns addressed**. Reviewers raised several points which were comprehensively addressed during the rebuttal:

- **Uplift baselines and statistical rigor (wohN, jAQG)**: We added standard deviations to all experiments in Table 1, and included direct query baselines to contextualize our results. New experiments in Table 3 (Appendix F.1) show that decomposition attacks yield significant uplift even against near-frontier open-weight models (Kimi K2, Qwen2.5-72B, Qwen3-235B, Llama-3.1-405B), showing that our findings are not artifacts of a weak baseline.

- **Defense details, false positives, and practicality (wohN, jAQG, fhmj)**: We added Table 5 (Appendix F.4) reporting precision and false positive rates at 90% and 99% recall for both pointwise and buffer defenses. The buffer defense substantially outperforms pointwise detection even under realistic conditions where harmful queries are rare relative to benign technical queries (WMDP).

- **Dataset release policy and dataset size (wohN, jAQG, fhmj)**: We added Appendix I containing the full questionnaire for dataset access. We have already shared BSD with multiple academic groups, frontier lab employees, and safety institutes, and have been responsive to reproduction requests. In terms of size, BSD’s questions yields thousands of decompositions for stateful evaluations and are comparable in scale to prior evaluations of similar difficulty. The pipeline can generate more and the current size only reflects budget constraints.

- **Multi-account evasion (jAQG, EwCY, fhmj)**: We clarified that while determined attackers could distribute queries across accounts, this imposes practical overhead (new emails, KYC-compliant payment methods). Notably, Anthropic's recent report on state-actor decomposition attacks [1] evidenced a setting where campaigns likely originated from single accounts, supporting the real-world viability of stateful defenses.

**Clarity and additions to the revised paper**. We added analysis on when decomposition beats jailbreaks. We also include sub-query compliance rates in Section 4.1. Beyond the experimental additions above, we incorporated our novel jailbreak+decomposition attack into the main text (Section 6). We believe these revisions substantively strengthen the paper's empirical foundation and clarity.

[1] https://www.anthropic.com/news/disrupting-AI-espionage

---

### Meta-Review · Area_Chair_44tZ · 2026-01-07

**Summary:**

The paper proposes a timely study of LLM misuse, illustrated by recent disclosures from (and about) big LLM providing companies.
The reviewers engaged with the authors, leading to an improved version of the manuscript, but several concerns remain valid as to wether the paper executes correctly what it promise, in particular, a stronger evaluation could improve the future versions of the paper, continuing on what the reviewers suggested.

**Reviewer Concerns:**

reviewers woN and jAQG concerns on statistical rigor led the authors to add standard deviations in table 1,
reviewers wohN, jAQG and fhmj's concerns on defence, false positives and practicality led the author to further precise FPRs
and reversers EwCY, fhmj and jAQG's concern on multi-account evasion helped the authors clarify what the overhead could be for the attackers (new emails, payment methods etc.),

**Reviewer Scores:**

EwCY might have raised from 2 to 4, though their skepticism towards the novelty and (more importantly) the evaluation used to assess model safety.

---

### Decision · Program_Chairs · 2026-01-26

Reject